# Microstructure Evolution and Dislocation Mechanism of a Third-Generation Single-Crystal Ni-Based Superalloy during Creep at 1170 °C

**DOI:** 10.3390/ma16145166

**Published:** 2023-07-22

**Authors:** Ruida Xu, Ying Li, Huichen Yu

**Affiliations:** Science and Technology on Advanced High Temperature Structure Materials Laboratory, AECC Key Laboratory of Aeronautical Materials Testing and Evaluation, Beijing Key Laboratory of Aeronautical Materials Testing and Evaluation, Beijing Institute of Aeronautical Materials, Beijing 100095, China; xjxxrd@163.com (R.X.); liying_patent@163.com (Y.L.)

**Keywords:** single-crystal Ni-based superalloy, creep, dislocation, deformation mechanism, γ/γ′ phases, topologically close-packed phase

## Abstract

The present study investigates the creep behavior and deformation mechanism of a third-generation single-crystal Ni-based superalloy at 1170 °C under a range of stress levels. Scanning electron microscopes (SEM) and transmission electron microscopes (TEM) were employed to observe the formation of a rafted γ′ phase, which exhibits a topologically close-packed (TCP) structure. The orientation relationship and elemental composition of the TCP phase and matrix were analyzed to discern their impact on the creep properties of the alloy. The primary deformation mechanism of the examined alloy was identified as dislocation slipping within the γ matrix, accompanied by the climbing of dislocations over the rafted γ′ phase during the initial stage of creep. In the later stages of creep, super-dislocations with Burgers vectors of a<010> and a/2<110> were observed to shear into the γ′ phase, originating from interfacial dislocation networks. Up to the fracture, the sequential activation of dislocation shearing in the primary and secondary slipping systems of the γ′ phase occurs. As a consequence of this alternating dislocation shearing, a twist deformation of the rafted γ′ phase ensued, ultimately contributing to the fracture mechanism observed in the alloy during creep.

## 1. Introduction

Single-crystal (SX) nickel-based superalloys are widely regarded as the primary material choice for turbine blades in aero-engines due to their exceptional mechanical properties, particularly at high temperatures [1,2]. As one of the main fracture modes for turbine blades at high temperatures, extensive research has been conducted to investigate the creep behavior of nickel-based superalloys in high-temperature environments [3,4,5,6,7,8]. During the initial creep stage, deformation is primarily governed by dislocation slipping in the γ matrix and climbing over the γ′ phase [8,9]. Then, the dislocations pile up at the interface of the γ phase and γ′ phase to form dislocation networks. Studies have demonstrated that the density of dislocation networks is inversely proportional to the minimum creep rate observed in third- and fourth-generation SX superalloys [10,11], which is influenced by alloying elements. To serve in harsher environments with elevated temperatures, the refractory elements are incorporated in the superalloys [7,12]. Notably, the addition of rhenium (Re), which predominantly dissolves in the γ matrix, acts as a solid solution strengthener and effectively slows down diffusion processes [13,14,15]. However, the presence of Re within the γ matrix increases lattice strain and enhances the solution-strengthening capacity of the γ matrix, impeding dislocation movement within the matrix [16]. Re has been reported to decelerate the creep strain rate during the steady-creep stage through its interaction with interfacial dislocations [17]. In the tertiary creep stage, dislocations shear into the γ′ phase in the form of super-dislocations, partials with anti-phase boundaries (APBs), and stacking faults [18,19]. Tian et al. [6] have shown that the inclusion of Re can reduce the stacking fault energy of the alloy, thereby enhancing its resistance to creep.

Meanwhile, the creep properties of SX nickel-based superalloys exhibit a close correlation with microstructure evolution. At elevated temperatures, the shape of the γ′ phase undergoes a transformation, resulting in the formation of an N-type or P-type rafted structure during the initial creep stage [20,21,22]. The transformation is influenced by the lattice misfit. The formation of an N-type rafted structure occurs with the negative lattice misfit between the γ phase and γ′ phase, whereas the positive lattice misfit would lead to a P-type rafted structure. When the dislocations shear, bypass, and climb over the γ′ phase, they are influenced by the microstructure degradation [23]. This microstructural degradation leads to reduced resistance to creep and a decreased creep lifetime [24,25]. However, the addition of rhenium (Re) can significantly impede the growth of the γ′ phase and contribute to microstructural stability [26]. It is worth noting that previous investigations into the deformation mechanisms of SX superalloys during creep have mostly focused on temperatures below 1100 °C [27]. The behavior and deformation mechanisms during creep at temperatures exceeding 1100 °C have been rarely explored, despite the fact that Re addition at these temperatures promotes the precipitation of topologically close-packed (TCP) phases [28,29,30]. The growth of TCP phases causes the depletion of solid solution-strengthening elements and consequently leads to a deterioration in mechanical properties [28,31]. The presence of refractory elements, as a consequence of high-temperature exposure or creep, contributes to the segregation, leading to the formation of TCP phases. The dislocation movement is also influenced by the occurrence of the TCP phase [32]. 

Hence, the purpose of this study is to investigate the creep behavior of an SX nickel-based superalloy at 1170 °C within a certain range of applied stresses. The creep characteristics, as microstructure evolution and the movement of dislocations in γ and γ′ phases, are investigated in detail. The deformation mechanism of the superalloy is discussed and determined.

## 2. Materials and Methods

For the present investigation, a third-generation single-crystal (SX) nickel-based superalloy was employed. The alloy composition includes a total of 20.5 wt.% of refractory elements, such as tungsten (W), molybdenum (Mo), tantalum (Ta), rhenium (Re), and niobium (Nb), where the percentage of Re is 4.5 wt.% [12]. The specific chemical compositions can be found in Table 1. The superalloy was directionally solidified to form [001]-orientated single-crystal bars in a vacuum furnace. Only single-crystal bars with a maximum deviation of 5° from the [001] orientation were selected for subsequent creep experiments, utilizing Laue-back reflection techniques. The heat treatment of the alloy involved a sequential process of 1613 K/6 h/air cooling + 1393 K/4 h/air cooling + 1143 K/32 h/air cooling. This treatment was carefully designed to control the morphology and volume of γ′ precipitates.

Following the heat treatment, the single-crystal bars were machined into specimens for creep testing. The schematic of the samples is illustrated in Figure 1, wherein the gage segment of the specimens possessed a length of 51.6 mm and a diameter of 10 mm. Creep tests at 1170 °C were conducted in accordance with ASTM E139-11 standards [33], applying tensile loads of 100 MPa. The creep test machine is an RD-100 Creep/Rupture Test Machine made by Changchunkexin Test Instrument Co., Ltd., Changchun, China. Additionally, three interrupted creep tests were performed under the conditions of 1170 °C/100 MPa after certain creep durations of 2 h, 10 h, and 50 h, respectively. A Quanta FEG 450 SEM microscope (FEI, Hillsboro, OR, USA) was employed for the observation of the microstructure evolution of the alloy. For transmission electron microscopy (TEM) observations, specimens were sliced from the middle portion of the gage section of the creep specimens and oriented along the [001] and [011] crystallographic directions. Before TEM analysis, these specimens underwent grinding with metallographic sandpaper to a thickness of 50 μm, followed by twin-jet electropolishing utilizing an electrolyte consisting of 10 wt.% perchloric acid and 90 wt.% absolute ethyl alcohol. TEM investigations were performed using a JEM-2010F microscope, operated at an accelerating voltage of 200 kV.

## 3. Results

### 3.1. Creep Behaviors of the Alloy

Figure 2 presents the creep curves and the strain-rate vs. time curves of the investigated alloy, while Table 2 provides a comprehensive overview of the quantitative creep characteristics exhibited by the alloy. The creep deformation behavior can be deduced from the creep curves as follows: (i) The creep curves display typical features corresponding to three creep stages: the initial creep stage, steady-creep stage, and tertiary creep stage. Notably, the creep strain rates exhibit a noticeable decrease during the onset of creep. Subsequently, the alloy enters the steady-creep stage, which constitutes the most significant portion of its overall life under a stress level of 100 MPa. When the creep enters the tertiary creep stage, the creep strain rate increases immediately; (ii) The minimum strain rate observed during the steady-creep stage under a stress level of 100 MPa measures at 0.122 $\times \;10^{-7}{\;s}^{-1}$.

Figure 3a displays the microstructure of the alloy after undergoing full heat treatment. Within the γ matrix, the cubic γ′ phase is uniformly distributed, with an average particle size of 410 nm and an average width of 65 nm for the γ matrix channels. The measured volume fraction of the γ′ phase amounts to approximately 68 vol.%. Subsequently, Figure 3b–d illustrate the microstructures after creeping for 2 h, 10 h, and up to fracture under conditions of 1170 °C/100 MPa. After 2 h of creep, the cubic γ′ phase undergoes a transformation into an ellipsoidal shape, while certain portions of the γ′ phase connect along the direction perpendicular to the applied stress, transforming into an N-type rafted structure. Moreover, the width of the horizontal γ matrix channel significantly increases, as evident in Figure 3b. These observations indicate the occurrence and completion of the rafting process at 1170 °C during the initial stages of creep. As the creep duration extends to 10 h (Figure 3c), the width of the horizontal γ matrix channel continues to grow, while the vertical γ matrix channel experiences degeneration and even disappearance. After fracture at 1170°C/100 MPa, a topological inversion phenomenon occurs, wherein the continuous γ channels isolating the γ′ phase develop into continuous γ′ rafts with an isolated γ phase. Furthermore, the γ′ phase exhibits a twisted morphology due to large plastic deformation. The presence of TCP phases is observed, as indicated by black arrows.

### 3.2. Analysis of Dislocation Configuration

The TEM images presented in Figure 4 depict the microstructural evolution of the alloy subjected to various durations of creep at 1170 °C/100 MPa. The plane of observation for the specimens aligns parallel to the (101) crystallographic plane. Figure 4a displays an illustration of the alloy’s morphology after a creep duration of 2 h. It reveals that the γ′ phase displays a rafted structure, while interfaces between the rafted γ and γ′ phases exhibit dislocation networks. The accumulation of dislocations within the γ matrix at the γ/γ′ interface leads to the gradual formation of denser dislocation networks, resulting in localized stress concentration. This stress concentration mechanism promotes the dislocation shear into the γ′ phase. Notably, upon subjecting the alloy to 10 h of creep at 1170 °C/100 MPa, the γ′ phase exhibits complete rafting perpendicular to the applied stress axis, as depicted in Figure 4b. The presence of a super-dislocation, indicated by arrows, is observed to shear into the γ′ phase. The morphology of the alloy approaching fracture is demonstrated in Figure 4c. At this critical stage, an abundance of dislocation arrays is observed within the γ′ phase, leading to a significant increase in dislocation density within the γ′ phase.

Figure 5 presents the dislocation configuration of the alloy after 2 h of creep. The dislocations primarily pile up at the interface between the γ and γ′ phases, and a super-dislocation, referred to as dislocation A, is observed shearing into the γ′ phase from the γ/γ′ interface. Dislocation A exhibits contrast when the diffraction vectors **g** = [200] and **g** = [$\bar{1}1\bar{1}$] are considered, while it disappears when the diffraction vectors **g** = [$02\bar{2}$] and **g** = [$11\bar{1}$] are considered. Based on the invisible criteria b·g=0 or ±2/3 () for dislocations, dislocation A is identified as having a Burgers vector $b_{A}=g_{02\bar{2}}\times g_{11\bar{1}}=\left[011\right]$. The trace vector of dislocation A is $[21\bar{1}$], and the slipping plane is determined as $\left(\bar{1}1\bar{1}\right)$ based on b×μ. Therefore, dislocation A is characterized as the [011] super-dislocation shearing into the γ′ phase. The magnified view of the interfacial dislocations is shown in Figure 5c, revealing dislocations with jog features indicated by the dashed lines. This suggests that the dislocations may traverse the rafted γ′ phase by climbing along the jogs.

Figure 6 presents the dislocation configuration within the γ′ phase of the alloy after 10 h of creep at 1170 °C/100 MPa. Dislocation networks are distributed at the interface between the γ and γ′ phases. The dislocations within the γ′ phase are labeled as B, C, and D, respectively. Dislocation B displays contrast when the diffraction vectors are **g** = [$1\bar{1}\bar{1}$] and **g** = [$1\bar{1}1$], whereas its contrast disappears at **g** = $[00\bar{2}]$ and **g** = [$\bar{2}20$]. Therefore, dislocation B is identified as having a Burgers vector $b_{B}=[110]$. The trace vector of dislocation B is $\mu_{B}=[1\bar{1}\bar{2}]$, and its slipping plane is identified as ($\bar{1}1\bar{1}$) based on b×μ. Dislocation C exhibits contrast when **g** = [$00\bar{2}$] and **g** = [$1\bar{1}\bar{1}$] are considered, while its contrast disappears at **g** = [$1\bar{1}1$] and **g** = [$\bar{2}20$]. Consequently, dislocation C is identified as having a Burgers vector $b_{C}=[110]$. The slipping plane of dislocation C is determined as ($1\bar{1}\bar{1}$), and its trace vector is [$1\bar{1}2$]. Dislocation D loses contrast at **g** = [$1\bar{1}1$] but displays contrast at **g** = [$00\bar{2}$], **g** = [$\bar{2}20$], and **g** = [$1\bar{1}\bar{1}$]. Dislocation D is identified as having a Burgers vector $b_{D}=\left[011\right]$. The trace vector of dislocation D is $\mu_{D}=\bar{2}20$, and its slipping plane is determined as ($\bar{1}\bar{1}1$).

Additionally, another dislocation with a zigzag character in the γ′ phase, marked as dislocation E, is observed originating from the γ/γ′ interface and may be influenced by adjacent dislocation networks. Dislocation E disappears in contrast at **g** = [$00\bar{2}$] but exhibits contrast at **g** = [$1\bar{1}1$], **g** = [$\bar{2}20$], and **g** = [$1\bar{1}\bar{1}$]. Accordingly, dislocation E is identified as a[010] super-dislocation according to the invisible criteria. The trace vector of dislocation E is [$\bar{2}00$], leading to the identification of its slipping plane as ($00\bar{1}$), based on b×μ.

In summary, the dominant deformation feature under the condition of 1170 °C is characterized by dislocation networks in the γ matrix and super-dislocations shearing into the γ′ phase, as observed from the analysis presented above.

## 4. Discussion

### 4.1. Analysis of Microstructure Evolution during Creep

During the creep process at 1170 °C, the microstructure of the alloy undergoes significant changes, characterized by the phenomena of rafting and coarsening. These transformations are influenced by both the lattice misfit and the applied stress. Specifically, when an alloy with a negative lattice misfit is subjected to tensile stress, its microstructure can transform into an N-type rafted structure. The formation of this structure is achieved through the process of orientational diffusion, wherein elements that promote the formation of the γ phase (referred to as γ-forming elements) diffuse horizontally in the direction of the applied stress, while elements that contribute to the formation of the γ′ phase (referred to as γ′-forming elements) diffuse vertically. Consequently, adjacent γ′ phases become connected, resulting in the development of a rafted structure. The phenomenon of topological inversion, as observed in Figure 3d, where the γ matrix is surrounded by the γ′ phase, has also been documented in Ref. [19].

Figure 7a illustrates the morphology of the TCP phase in the alloy subsequent to fracture at 1170 °C and under a stress of 100 MPa. The TCP phase appears in a needle-like form and is identified as the σ phase, which assumes a lamellar structure within the alloy. Figure 7b presents the diffraction pattern associated with the TCP phase. The crystallographic relationship is acquired as ${\left[010\right]}_{\gamma /\gamma^{\prime }}∥{\left[100\right]}_{\sigma }$. Typically, the lamellar σ phase aligns parallel to the {111} planes of the γ matrix. The angle between the σ phase and the γ′ phase is approximately 45°, as depicted in Figure 3b,c. The reported orientation relationship between the σ phase and the γ phase is ${\left(00\bar{1}\right)}_{\sigma }∥{\left(1\bar{1}1\right)}_{\gamma }$ [24]. The habit plane for the precipitation of the σ phase is the {111} planes of the γ matrix, coinciding with the slipping planes of dislocations within the γ matrix. The presence of the lamellar σ phase hinders dislocation movement by impeding their climb or shear into the σ phase. Consequently, dislocations pile up at the interface between the σ phase and the γ matrix, ultimately leading to crack nucleation.

Figure 8 presents a high-angle annular dark-field (HAADF) image and corresponding elemental distribution maps of the region encompassing the TCP phase. The chemical compositions of the TCP phase, γ matrix, and γ′ phase in Figure 8 are measured and listed in Table 3. Notably, the TCP phase exhibits element segregation, particularly with respect to elements such as W, Cr, and Re, while elements such as Ni, Al, and Co are present in lower concentrations within the TCP phase. This phenomenon of element segregation is attributed to the presence of γ-forming elements in the TCP phase surrounding the γ′ phase. The segregation of refractory elements like Re, W, and Cr within the TCP phase serves to weaken the alloy’s solution-strengthening characteristics.

In summary, during the creep process at 1170 °C, the alloy’s microstructure undergoes rafting and coarsening, influenced by both the lattice misfit and the applied stress. Tensile stress induces the formation of an N-type rafted structure through orientational diffusion, leading to the connection of adjacent γ′ phases. The presence of the TCP phase, identified as the σ phase, with its lamellar structure and its alignment along the {111} planes of the γ matrix, hampers dislocation movement, eventually resulting in crack nucleation. Furthermore, the TCP phase exhibits element segregation, particularly involving refractory elements, thereby weakening the alloy’s solution-strengthening properties.

### 4.2. Deformation Mechanisms of Creep

During the initial creep stage, as the γ′ phase transforms into an N-type rafted structure, dislocations within the γ matrix become activated and pile up at the interface of the γ/γ′ phase. This phenomenon is clearly depicted in Figure 5 and Figure 6, where most dislocations slip within the γ matrix and form immobile interfacial dislocation networks. However, mobile dislocations that reach the interface can interact with the dislocation network, altering their slipping direction and facilitating dislocation climbing over the rafted γ′ phase. The presence of jog features in the interfacial dislocations, as observed in Figure 5, further supports the idea that dislocations can traverse the γ′ phase by moving along jogs. A two-dimensional, simplified schematic representation of dislocation climbing is provided in Figure 9.

The process of dislocation climbing is influenced by both temperature and applied stress. The critical stress ($\sigma_{i}$) required for dislocation climbing at elevated temperatures can be described by Equation (1) [16]:(1)$$\sigma_{i}=\frac{G\cdot b}{\left[8\pi \left(1-\nu \right)h\right]\mathrm{kT}}$$

Here, *G* represents the shear modulus, *b* represents the magnitude of the Burgers vector, *ν* represents the Poisson’s ratio, *h* represents the height of the γ′ phase, and *T* represents the temperature. In Equation (1), it is observed that the critical stress for the dislocation climb is influenced by the shear modulus (*G*), the height of the rafted γ′ phase (*h*), and the temperature (*T*). Equation (1) specifically emphasizes the inverse relationship between the critical stress ($\sigma_{i}$) and temperature (*T*). Furthermore, the shear modulus (*G*) and the volume fraction and size of the γ′ phase are affected by temperature. As the temperature increases, the alloy’s shear modulus decreases accordingly. Similarly, the height of the γ′ phase is also influenced by temperature, manifesting primarily through the rafting and coarsening of the microstructure. For SX Ni-based alloys, the rafting and coarsening of the γ′ phase typically occur above 1000 °C, leading to an increase in the size of the γ′ phase. Therefore, due to the combined influence of these three factors, compared to creep at lower temperatures, the critical stress for dislocation climbing decreases with increasing temperature, making dislocation climbing more likely to occur at higher temperatures. In the creep deformation at 1170 °C, the climbing process is regarded as the dominant mechanism governing the plastic strain during creep [34].

As creep progresses, the stress concentration resulting from the dense dislocation networks within the γ matrix promotes the shearing of dislocations into the γ′ phase. The dislocations within the γ′ phase primarily manifest as a/2<110> super-dislocations, as depicted by dislocation A in Figure 5. These dislocations in the γ′ phase may undergo decomposition, leading to a configuration comprising two a/3<112> Shockley partial dislocations separated by stacking faults [17]. Figure 10 illustrates a set of dislocation partials situated within the {111} slipping plane. The magnification of the dashed-square area in Figure 10a shows the stacking fault ribbon appearing as a parallel fringe pattern clear, as displayed in Figure 10b.This particular dislocation structure arises from the decomposition of the a/2<110> super-dislocation within the γ′ phase, resulting in partial dislocations accumulating at obstacles. Furthermore, a zigzag-shaped a<010> super-dislocation is observed within the γ′ phase, represented by dislocation E in Figure 6. As the a<010> super-dislocation slips along the ($00\bar{1}$) planes, its Schmid factor is determined to be zero when the applied stress aligns with the [001] direction. Consequently, the dislocation remains immobile and can only move through climbing within the γ′ phase. The formation of the a<010> super-dislocation can be reasonably attributed to the following reaction at the γ/γ′ interface:(2)$${\frac{a}{2}\left\langle 110\right\rangle }_{\gamma }+{\frac{a}{2}\left\langle \bar{1}10\right\rangle }_{\gamma }\to {a\;\left\langle 010\right\rangle }_{\gamma^{\prime }}$$

In the later stages of creep, dislocations from different <111> slip systems alternately shear into the rafted γ′ phase. This alternating activation of primary and secondary slipping systems results in the twisting of the rafted γ′ phase, as observed in Figure 3d. The degree of twist in the rafted γ′ phase increases the strain experienced by the alloy during creep, ultimately leading to fracture.

## 5. Conclusions

In summary, the investigation clarified the creep behavior and deformation mechanism of the third-generation single-crystal Ni-based superalloy at an elevated temperature. The observation of the rafted γ′ phase and its interaction with the γ matrix, along with the identification of dislocation slipping and shearing as the principal deformation mechanisms, provides valuable insights into the alloy’s mechanical properties under creep conditions.

(1)During creep, the initially cubic γ′ phases undergo a transformation into a rafted morphology through thickening and coarsening. Concurrently, the presence of the TCP phase is observed at various stages of creep at a temperature of 1170 °C. Through HAADF-EDS mapping, it is revealed that elements such as Re, W, and Cr exhibit significant segregation within the TCP phase. This selective distribution of refractory elements within the TCP phase has the detrimental effect of weakening the alloy’s solution-strengthening mechanism and consequently diminishing its creep performance.(2)In the initial creep stage at 1170 °C, the primary mechanism of deformation in the alloy is identified as dislocation slipping within the γ matrix, accompanied by the process of dislocation climbing over the rafted γ′ phase.(3)As the creep progresses to later stages, super-dislocations characterized by a Burgers vector of a<010> and a/2<110> shear into the γ′ phase, originating from interfacial dislocation networks. These networks not only serve as a source of dislocations for the γ′ phase but also impede the movement of dislocations within the γ phase.

## Figures and Tables

**Figure 1 materials-16-05166-f001:**
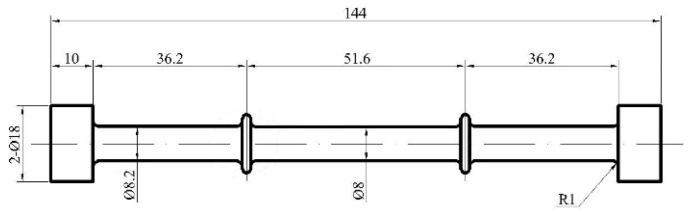
Schematic diagram of the creep specimen (unit: mm).

**Figure 2 materials-16-05166-f002:**
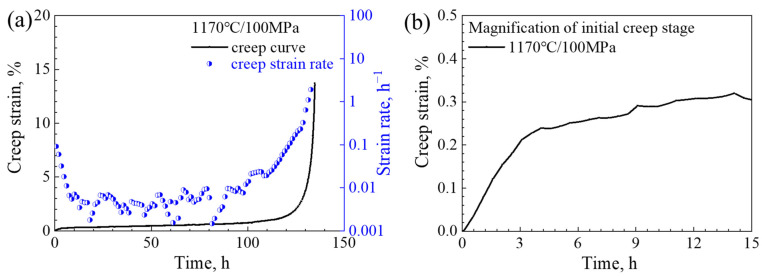
Creep properties of the alloy at 1170 °C. (**a**) Overall creep strain curves and strain-rate vs. time curves; (**b**) magnification of the initial creep stage.

**Figure 3 materials-16-05166-f003:**
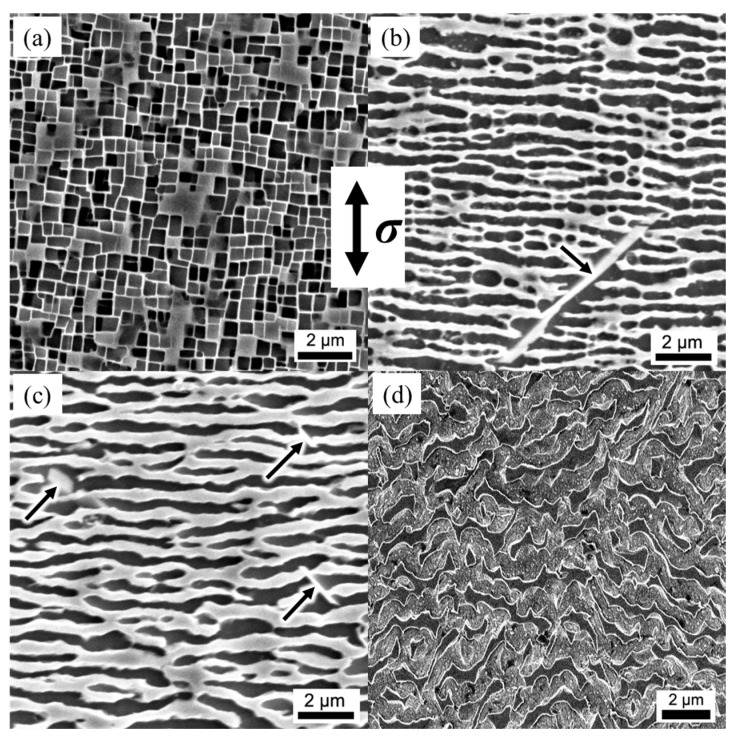
Microstructure evolution of the alloy during creep at 1170 °C and 100 MPa. (**a**) After full heat treatment; (**b**) creeping for 2 h; (**c**) creeping for 10 h; (**d**) failure.

**Figure 4 materials-16-05166-f004:**
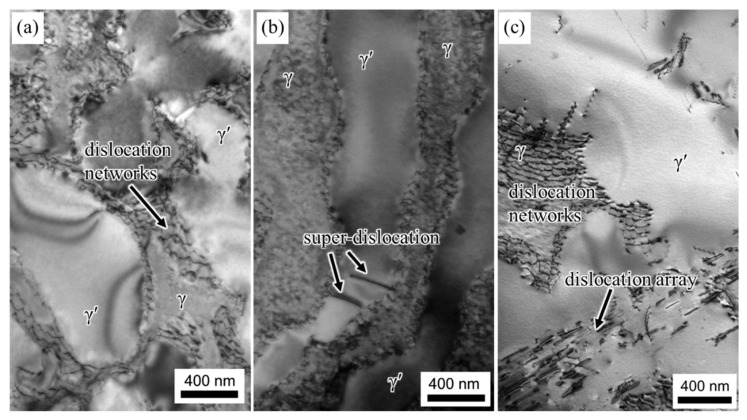
Microstructures after the alloy crept for different durations: (**a**) 2 h; (**b**) 10 h; (**c**) 135 h (up to fracture) at 1170 °C/100 MPa.

**Figure 5 materials-16-05166-f005:**
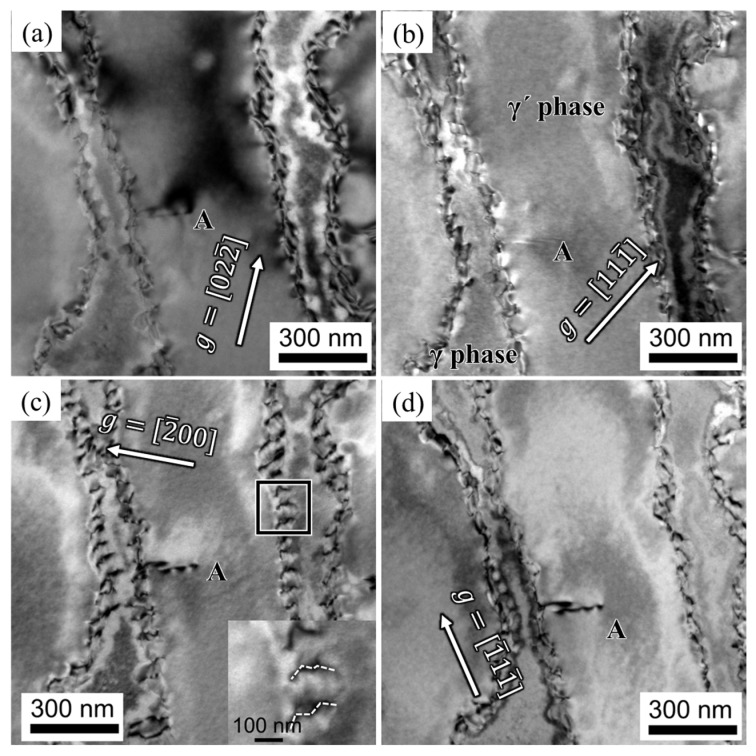
Dislocation configurations of the alloy after creeping for 2 h at 1170 °C/100 MPa: (**a**) $g=[02\bar{2}]$, (**b**) $g=[11\bar{1}]$, (**c**) $g=[\bar{2}00]$, (**d**) $g=[\bar{1}1\bar{1}]$.

**Figure 6 materials-16-05166-f006:**
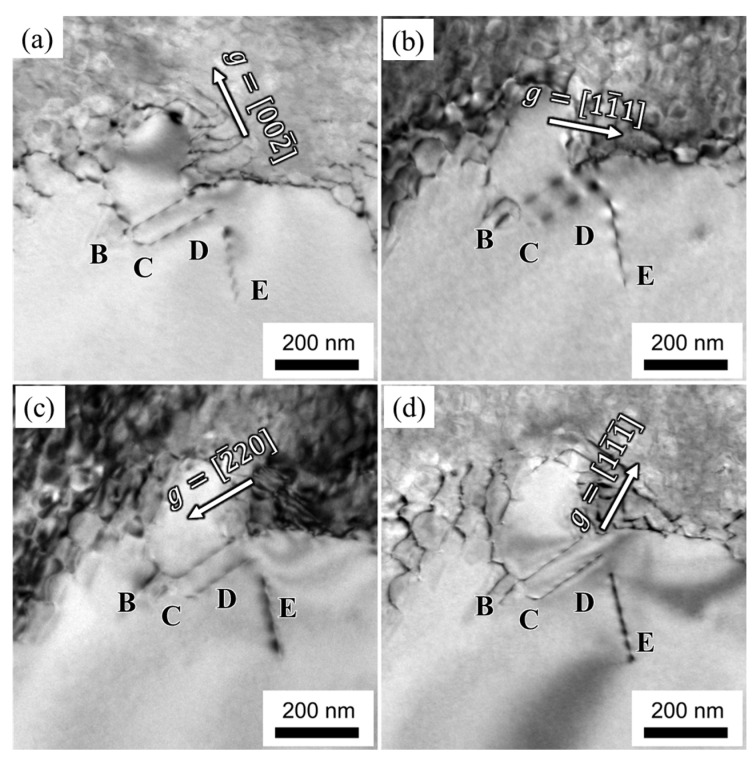
Dislocation configurations of the alloy after creeping for 10 h at 1170 °C/100 MPa: (**a**) $g=[00\bar{2}]$, (**b**) $g=[1\bar{1}1]$, (**c**) $g=[\bar{2}20]$, (**d**) $g=[1\bar{1}\bar{1}]$.

**Figure 7 materials-16-05166-f007:**
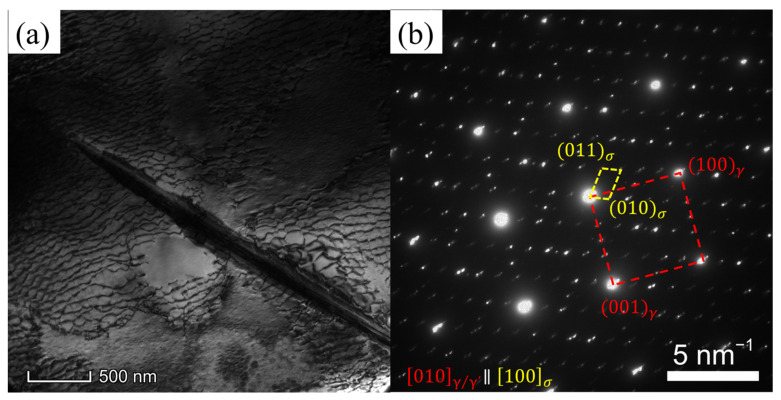
(**a**) TEM image of the TCP phase after fracturing at 1170 °C/100 MPa, (**b**) the diffraction pattern of the TCP phase in (**a**).

**Figure 8 materials-16-05166-f008:**
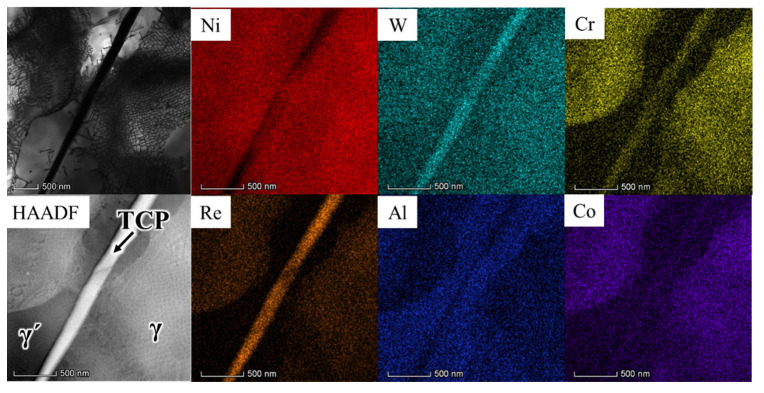
The bright TEM image, HAADF image, and EDS elemental distribution maps of the TCP phase.

**Figure 9 materials-16-05166-f009:**
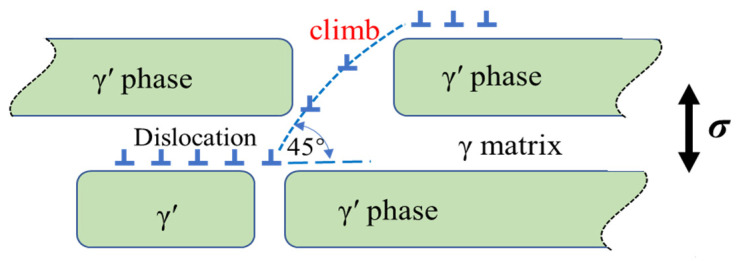
The simplified schematic of dislocation climbing over the rafted γ′ phase.

**Figure 10 materials-16-05166-f010:**
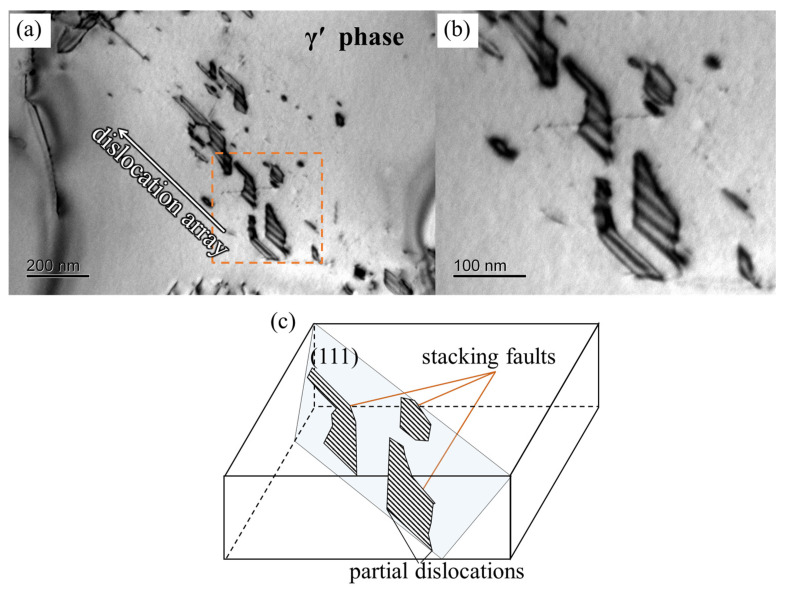
Dislocation arrays pile up in the γ′ phase of the alloy after creeping for 50 h at 1170 °C/100 MPa: (**a**) the TEM morphology; (**b**) magnification of the dashed-square area in (**a**); (**c**) the schematic diagram of stacking faults.

**Table 1 materials-16-05166-t001:** Chemical composition of the alloy used in this study (wt.%) [12].

Cr	Co	Mo	W	Ta	Re	Nb	Al	Hf	C	Y	Ni
3.5	7	2	6.5	7.5	4.5	0.5	5.6	0.1	0.008	0.001	Bal.

**Table 2 materials-16-05166-t002:** Quantitative creep features of the alloy at 1170 °C/100 MPa.

	Creep Life/h	Duration of Steady-Creep Stage/h	$\mathrm{Minimum}\;\mathrm{Strain}\;\mathrm{Rate}/\times 10^{-7}s^{-1}$
Stress/MPa	134.9	92.9	0.122

**Table 3 materials-16-05166-t003:** Chemical composition of the γ matrix, γ′ precipitates, and TCP phase in Figure 8 (atomic fraction, %).

	Al	Cr	Co	Ni	W	Re
γ matrix	6.72	8.01	9.94	65.13	3.2	3.05
γ′ phase	14.83	1.33	5.92	69.46	2.84	0.35
TCP phase	5.03	6.59	7.12	53.37	8.52	16.05

## Data Availability

The data presented in this study are available on request from the corresponding author.

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
