# Peer review of "Microstructure Evolution and Dislocation Mechanism of a Third-Generation Single-Crystal Ni-Based Superalloy during Creep at 1170 °C"

_materials, 2023, doi:10.3390/ma16145166_

Round 1

Reviewer 1 Report

Comments and suggestions to improve the article are in the attached file

Author Response

Reviewer 1: The authors investigated the creeping behavior and deformation mechanism of the third generation single-crystalline Ni superalloy at 1170°C under various stress levels. TEM analysis was used to observe the γ' phase and its interaction with the γ matrix, as well as to identify dislocation slip and shear as the main deformation mechanisms, providing valuable information for understanding the mechanical properties of alloys under creep conditions.

Response: We are grateful for the reviewer’s critical comments and thoughtful suggestions for our manuscript. According to these comments and suggestions, we have carefully made modifications to the original manuscript. Those comments are helpful for authors to revise and improve our paper. We have studied the comments carefully and tried our best to revise the manuscript according to the comments from reviewers and editors. A Revised Manuscript file having modification marks also has been submitted. The new supplements and corrections to the text according to the reviewers’ comments are highlighted in yellow mark.

There are some small ambiguities in the text that can be corrected after a minor review of the article, which is my recommendation for this publication.

Suggestions to improve the paper:

  1. Line 71. 2. Materials and Methods: Did the SX nickel-based superalloy used in the article have a trade name? Or was it developed by the authors? If this alloy has a name, please add it.

Response: The SX Nickel-based superalloy used in the article are developed by Jiarong Li of AECC Beijing Institute of Aeronautical Materials. Due to confidentiality requirements, the authors are not allowed to give its trade name in the manuscripts. We appreciate for your understanding and supports.

  1. Line 71. 2. Materials and Methods: In the abstract there is information about SEM analysis; however, there is no information about it in the article. Are all the figures in the article from a TEM microscope? If any are from an SEM microscope, then please add information about the SEM device to Materials and Methods. If not, then please remove the information about SEM from the abstract.

Response: Thanks for your careful reviews. We used the SEM microscope for the observation of Microstructure evolution of the alloy, as seen in Fig.3. And the SEM microscope is Quanta FEG 450 SEM microscope. We have added its information to the manuscript in Section 2.

  1. Line 85. You could consider adding information about the device on which the creep tests were performed. For now, there is only information that they were performed according to ASTM standard, which is correct but you could add the name and manufacturer of the device.

Response: The creep test machine is RD-100 Creep/Rupture Test Machine made by Changchunkexin Test Instrument Co., LTD. The related information has been also added to the manuscript in Section 2.

  1. Line 96. Table 1: The chemical composition of the alloy used in this study was determined by the authors or was taken from the literature. In the text, there is a reference to the chemical composition, while in Table 1 there is no reference. Please, add a reference from the literature or information on how the chemical composition of the alloy was measured.

Response:  We appreciate your identification of our shortcomings. In response, we have incorporated citations to the literature within Table 1.

  1. Line 328. Results: The literature review is admittedly extensive, but items from the publisher "mdpi" are missing. Please take into account the policy of indexing indicators of the publisher. Please review the noteworthy positions and add some of them to the article.

Response: Thanks for your reminding. After further research of the references, we revised the introduction and add the items of the publisher MDPI. We appreciate your support and understanding.

Reviewer 2 Report

Dear authors Attached please find some comments or suggestions.
Best regards.

Author Response

Best regards.

Reviewer 3 Report

In my opinion, this contribution should be considered for acceptance. The paper addresses a highly interesting and current topic, which holds significant relevance in multiple fields, particularly in aerospace industry applications where the creep phenomena play a crucial role. Superalloys, with their exceptional high-temperature mechanical properties, are extensively employed in the aerospace industry. The study of creep mechanisms in these materials is of utmost importance to ensure the structural integrity and reliability of components subjected to prolonged exposure to elevated temperatures. Moreover, the significance of understanding the creep behavior of superalloys extends beyond aerospace, as they find application in energy production, gas turbines, and nuclear power plants, where extreme environments with high temperatures, mechanical stress, and corrosion prevail. By comprehending the intricacies of creep mechanisms, researchers can develop advanced materials, predictive models, and design guidelines, contributing to enhanced performance, extended service life, and increased safety margins. Through theoretical and practical demonstrations, this manuscript successfully highlights the critical role of the γ' phase and its interaction with the γ matrix while identifying dislocation slipping and shearing as the principal deformation mechanisms. Moreover, the paper is well structured, well written, and clear about its significance and relevance.

Author Response

In my opinion, this contribution should be considered for acceptance. The paper addresses a highly interesting and current topic, which holds significant relevance in multiple fields, particularly in aerospace industry applications where the creep phenomena play a crucial role. Superalloys, with their exceptional high-temperature mechanical properties, are extensively employed in the aerospace industry. The study of creep mechanisms in these materials is of utmost importance to ensure the structural integrity and reliability of components subjected to prolonged exposure to elevated temperatures. Moreover, the significance of understanding the creep behavior of superalloys extends beyond aerospace, as they find application in energy production, gas turbines, and nuclear power plants, where extreme environments with high temperatures, mechanical stress, and corrosion prevail. By comprehending the intricacies of creep mechanisms, researchers can develop advanced materials, predictive models, and design guidelines, contributing to enhanced performance, extended service life, and increased safety margins. Through theoretical and practical demonstrations, this manuscript successfully highlights the critical role of the γ' phase and its interaction with the γ matrix while identifying dislocation slipping and shearing as the principal deformation mechanisms. Moreover, the paper is well structured, well written, and clear about its significance and relevance.

Response: We sincerely appreciate your review and support for our work. SX Ni-based alloys exhibit excellent mechanical properties and corrosion resistance at high temperatures, making them highly promising for various high-temperature applications, including but not limited to turbine blades. Over time, these alloys have undergone several generations of development, ranging from the first to the fifth generation. Among the second-generation SX Ni-based alloys, alloys such as CMSX-4, Rene N5, DD6, have already gained widespread application, while the third-generation single-crystal alloys are on the verge of being employed in engineering field. Our research elucidates the creep performance and deformation mechanisms of such alloys at ultra-high temperatures, with the aim of providing support for their life prediction and safety design considerations.

Reviewer 4 Report

the current paper is studying the Microstructure evolution and dislocation mechanism of a third- 2 generation Single Crystal Ni-based Superalloy during creep at 3 1170℃, the topic and results are interesting but I have concerns related to the results and analysis of the results as follows:

1. Fig. 4 dislocations are not visibly clear, a super dislocation was mentioned, what is a super dislocation?

2. Fig. 5 and 6 show dislocation study, this is improper study a complete TEM contrast analysis is needed.

3. there is a parallel diffraction in the SAED in Fig. 7(b)

4. Fig. 10 shows illustration of dislocation arrangement showing the presence of stacking faults however, there is not microstructure analysis confirming the presence of stacking faults. 

the quality of the english language is acceptable but it can be improved

Author Response

Best wishes. 

Round 2

Reviewer 2 Report

Dear authors

Thank you for considering my suggestions and modifications. To my understanding and in fact, your alloy exhibited a stress plateau (constant) at high stress. In other words, the fracture time in comparison with tests at 1100°C or higher (if you can find results above 1100°C in the open literature) remains almost independent of stress. I urge you to take a closer look at these life curves and this point. Indeed, these alloys should have an acceptable service life when the temperature is so high and when the stress becomes very high for safety requirements, particularly for moving blade applications. Best regards

Reviewer 4 Report

The authors have followed the suggested comments and done the appropriate changes

English is good